

# Will 1,2-dihydro-1,2-azaborine-based drugs resist metabolism by cytochrome P450 compound I?

Pedro J. Silva

FP-ENAS, Faculdade de Ciências da Saúde, Universidade Fernando Pessoa, Porto, Portugal

## ABSTRACT

1,2-dihydro-1,2-azaborine is a structural and electronic analogue of benzene which is able to occupy benzene-binding pockets in T4 lysozyme and has been proposed as suitable arene-mimicking group for biological and pharmaceutical applications. Its applicability in a biological context requires it to be able to resist modification by xenobiotic-degrading enzymes like the P450 cytochromes. Quantum chemical computations described in this work show that 1,2-dihydro-1,2-azaborine is much more prone to modification by these enzymes than benzene, unless steric crowding of the ring prevents it from reaching the active site, or otherwise only allows reaction at the less reactive $C_4$-position. This novel heterocyclic compound is therefore expected to be of limited usefulness as an aryl bioisostere.

## INTRODUCTION

1,2-dihydro-1,2-azaborine (abbreviated in this paper as "azaborine") is a structural and electronic analogue of benzene which, like benzene, undergoes classical electrophilic aromatic substitution (*Pan, Kampf & Ashe, 2007*) but, in contrast to benzene, also readily undergoes nucleophilic aromatic substitution under mild reaction conditions (*Lamm et al., 2011*). Computational studies have shown azaborines to be generally much more reactive towards one-electron oxidation and electrophilic substitution than their corresponding benzene analogues (*Silva & Ramos, 2009*). Azaborines are generally stable in water and react sluggishly with oxygen when substituted on their boron atoms with electron-withdrawing substituents (*Lamm & Liu, 2009*). These benzene isosteres are able to occupy benzene-binding pockets in T4 lysozyme (*Liu et al., 2009*) and have been proposed as suitable arene-mimicking groups for biological and pharmaceutical applications (*Marwitz et al., 2007*). Their deployment as useful components of drug scaffolds requires, however, that they are stable in the presence of drug-metabolizing enzymes such as the P450 cytochromes which hydroxylate the related benzene ring (*Guengerich, 2003*; *Guengerich, 2008*).

The active oxidant species of cytochrome P450 (Compound I) is a thiolate-bound heme compound which possesses two unpaired electrons in its Fe = O moiety and one unpaired electron delocalized throughout the porphyrin ring and the thiolate ligand (*Schöneboom et al., 2002* and references therein). Depending on the orientation of this lone spin relative

Corresponding author
Pedro J. Silva, pedros@ufp.edu.pt

to the Fe = O-localized spins, compound I may exist in a doublet ($S = 1/2$) or a quartet ($S = 3/2$) state, which have very similar energies (*Rydberg, Sigfridsson & Ryde, 2004* and references therein). Extensive experimental and computational investigations on the reaction of compound I towards benzene and other aromatic compounds (*Guroff et al., 1967*; *Jerina et al., 1968*; *Burka, Plucinski & Macdonald, 1983*; *Koop, Laethem & Schnier, 1989*; *Korzekwa, Swinney & Trager, 1989*; *Koerts et al., 1998*; *De Visser & Shaik, 2003*; *Bathelt et al., 2003*; *Bathelt, Mulholland & Harvey, 2008*) have shown that the initial formation of a $\sigma$-adduct between compound I and the aromatic compound is endergonic and that the subsequent formation of different products (arene oxides, phenols, or ketones) is ruled by a complex potential energy surface, which is sensitive to the reaction environment and to the mode of attack of the benzene (either perpendicular or parallel to the plane of the porphyrin ring). In this paper, we analyze the metabolic stability of 1,2-azaborines towards P450 enzymes through the computational investigation of their reactions with "compound I."

## COMPUTATIONAL METHODS

The geometries of every molecule described were optimized using B3LYP (*Lee, Yang & Parr, 1988*; *Becke, 1993*; *Hertwig & Koch, 1995*). Autogenerated delocalized coordinates (*Baker, Kessi & Delley, 1996*) were used in geometry optimizations performed with 6-31G(d) (*Ditchfield, Hehre & Pople, 1971*; *Hehre, Ditchfield & Pople, 1972*) for all elements except for Fe, which used the SBKJ VDZ (*Stevens et al., 1992*) basis set in combination with the SBKJ pseudo-potential (*Stevens et al., 1992*) for the inner shells corresponding to the (1s2s2p) core of Fe. Single-point energies of the DFT-optimized geometries were then calculated using the same functional using the 6-311 + G(2d,p) (*Hariharan & Pople, 1973*; *Krishnan et al., 1980*; *Clark et al., 1983*; *Frisch, Pople & Binkley, 1984*) basis set for all elements except Fe, which used the s6-31G* basis set, specifically developed by *Swart et al. (2010)* to afford more reliable spin-state splittings. Zero-point vibrational effects (ZPVE) were computed using a scaling factor of 0.9804 for the computed frequencies. Atomic charge and spin density distributions were calculated with a Mulliken population analysis (*Mulliken, 1955*) based on symmetrically orthogonalized orbitals (*Löwdin, 1970*). Geometries of products were obtained from those of the corresponding transition states upon slight deformation of the coordinate corresponding to the imaginary frequency, followed by unconstrained reoptimization. In the few instances where no transition state could be found, product geometries were obtained from extensive exploration of the potential energy surface using two-dimensional scans. All energy values described in the text include solvation effects ($\varepsilon = 10$) computed using the Polarizable Continuum Model (*Tomasi & Persico, 1994*; *Mennucci & Tomasi, 1997*; *Cossi et al., 1998*) implemented in Firefly. All computations were performed with the Firefly (*Granovsky, 2013*) quantum chemistry package, which is partially based on the GAMESS (US) (*Schmidt et al., 1993*) source code. Intra- and inter-molecular dispersion effects on the energies of the gas-phase B3LYP-optimized species were computed with the DFT-D3 formalism developed by *Grimme et al. (2010)*.

**Table 1 Energies (in kcal mol$^{-1}$, *vs.* the reactant state) of the transition states ($^2$TS and $^4$TS) and products ($^2$product and $^4$product) of direct attack benzene by compound I.** Species preceded by $^2$ are in the doublet ($S = 1/2$) state, whereas those preceded by $^4$ are in the quartet state ($S = 3/2$). These values cannot be directly compared to the experimental barriers due to the neglect of vibrational/rotational/translational contributions to entropy. Inclusion of entropic effects increases barriers by 4–6.5 kcal mol$^{-1}$ due to the loss of vibrational entropy in the transition state (see Supplemental Information).

| Level of theory | $^2$TS | $^2$Product | $^4$TS | $^4$Product | Reference |
|---|---|---|---|---|---|
| B3LYP ($\varepsilon = 5.7$) | 17.5–18.1 | 12.3–13.5 | 20.6 | 14.0 | *De Visser & Shaik (2003)* |
| B3LYP ($\varepsilon = 4.0$) | 15.6–17.9 | 6.1–6.9 | n.d | n.d | *Bathelt et al. (2004)* |
| B3LYP (gas phase only, including ZPVE) | 20.7 | n.d. | 21.1 | n.d. | *Rydberg, Ryde & Olsen (2008)* |
| QM/MM B3LYP/CHARMM27 | 20.4 | n.d. | 20.4 | n.d. | *Lonsdale, Harvey & Mulholland (2012)* |
| QM/MM B3LYP-D2/CHARMM27 | 13.5 | n.d. | 11.9 | n.d. | *Lonsdale, Harvey & Mulholland (2012)* |
| PBE0 (gas phase only, no ZPVE) | 18.8 | 8.8 | 24.4 | n.d. | *Tomberg et al. (2015)* |
| B3LYP-D3//B3LYP ($\varepsilon = 10.0$) (including ZPVE) parallel attack | **16.1** | **7.6** | **21.6** | **7.9** | This work |
| B3LYP-D3//B3LYP ($\varepsilon = 10.0$) (including ZPVE) perpendicular attack | **16.9** | **9.4** | **16.9** | **5.9** | This work |

## RESULTS

The experimental rates of benzene hydroxylation by the thiolate-bound compound I present in cytochrome P450 and haloperoxydases range from 4.6 min$^{-1}$ (*Koop, Laethem & Schnier, 1989*) to 8 s$^{-1}$ (*Karich et al., 2013*), which translate to activation free energies from 16.9 kcal mol$^{-1}$ to 19.8 kcal mol$^{-1}$. The computationally-derived activation energies vary from 12 kcal mol$^{-1}$ to 21 kcal mol$^{-1}$, depending on the theory level, model size, and inclusion (or not) of ZPVE, dispersion effects, or solvation (Table 1). Analysis of the susceptibility of 1,2-dihydro-1,2-azaborine to attack by compound I therefore required us to start our investigation by determining the influence of our theory level on the energetic barrier of the analogous reaction of benzene.

In the doublet potential energy surface (Fig. 1), we observed that the electronic structure of the reaction product depends on the aryl mode of attack: when benzene approaches the doublet state of compound I perpendicularly to the porphyrin ring ("side-on" in the nomenclature of *Bathelt et al., 2004*), half an electron is transferred from the benzene to the Fe ligands (porphyrin and thiolate) with concomitant spin rearrangements, which lead to the loss of one spin from the Fe–O moiety , mostly to the thiolate ligand (0.52 spin) and substrate (0.32 spin). In contrast, a parallel mode of attack ("face-on" in the nomenclature of *Bathelt et al., 2004*) yields the transfer of almost a full spin (0.86) (but no charge) from the thiolate and porphyrin to the benzene. These results are similar to the observation of a cation-like and a radical-like adduct by *Bathelt et al. (2004)*, though these workers were able (unlike us) to find both adducts with either attack mode.

Without taking into account zero-point vibrational effects, the quartet state of compound I lies only 0.4 kcal mol$^{-1}$ above the doublet state, and the quartet portential energy surface is therefore very accessible. In this spin state, no dramatic differences in electronic structure were found between both attack modes, which always yield a radical-like adduct on the benzene. In the perpendicular attack mode, the quartet state has the same energetic barrier

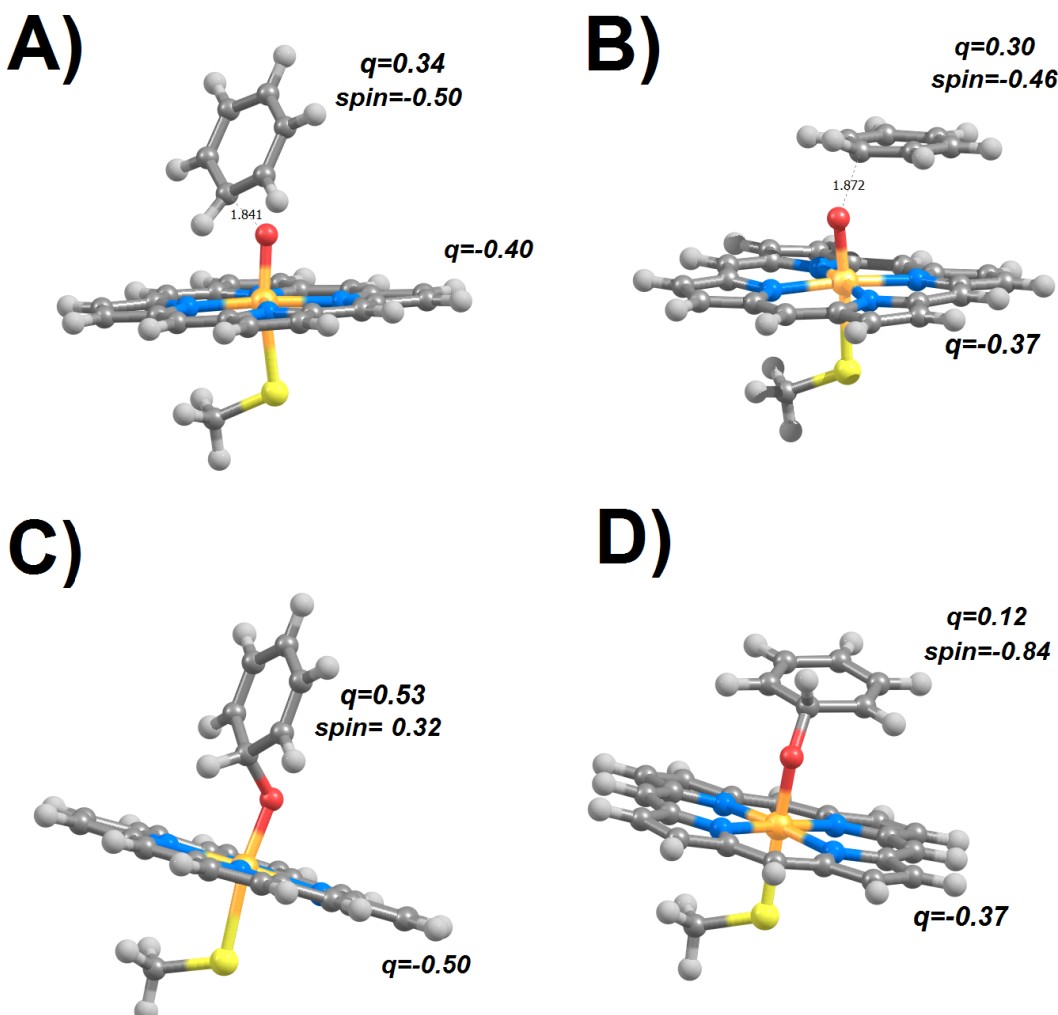

**Figure 1** **Transition states (A and B) and products (C and D) arising from perpendicular (A and C) or parallel (B and D) attack of benzene by compound I in the doublet ($S = 1/2$) state.** Charges (and spins) on the substrate moiety and on the porphyrin ring are highlighted.

as the doublet state, but produces a more stable product. Such a competitive benzene hydroxylation in the quartet state has not been found by earlier workers, whose studies on the subsequent rearrangement of the compound I/benzene adduct to yield phenol, ketone or epoxide (*Bathelt, Mulholland & Harvey, 2008*) focused only on the doublet surface due to the higher activation energies they observed for the formation of the compound I/benzene adduct in the quartet state.

The energy of the reactant state of compound I towards benzene is mostly independent of the spin state of compound I and of the parallel/perpendicular orientation of benzene. In contrast, the perpendicular orientation of 1,2-dihydro-1,2-azaborine is almost 8 kcal mol$^{-1}$ more favorable than the parallel orientation, due to the stabilization provided by hydrogen binding between the nitrogen-bound hydrogen and the compound I oxygen in the perpendicular orientation. This difference is not, by any means, the most dramatic

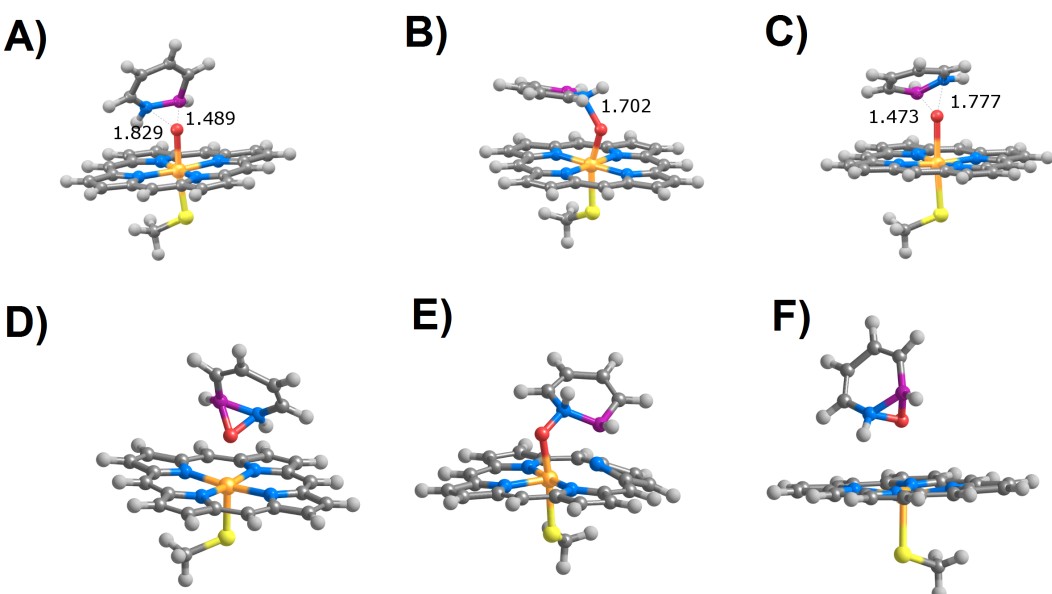

**Figure 2 Transition states (A–C) and products (D–F) arising from attack of the nitrogen atom in aza-borine by compound I.** (A and D) $S = 1/2$, perpendicular attack; (B and E) $S = 1/2$, parallel attack; (C and F) $S = 3/2$.

**Table 2 Energies (in kcal mol$^{-1}$, *vs.* the most stable reactant state) of the transition states ($^2$TS and $^4$TS) and products ($^2$product and $^4$product) for the direct attack of the heteroatoms in 1,2-dihydro-1,2-azaborine by compound I.** Species preceded by $^2$ are in the doublet ($S = 1/2$) state, whereas those preceded by $^4$ are in the quartet state ($S = 3/2$). All energy values include solvation effects ($\varepsilon = 10.0$), zero-point vibrational energy and dispersion effects. Transition states with activation energies above the activation energy of the reaction of compound I towards benzene are highlighted in bold. Unless otherwise noted, all products are $\sigma$-adducts of the substrate.

|  | $^2$TS | $^2$Product | $^4$TS | $^4$Product |
|---|---|---|---|---|
| N (parallel orientation) | **33.4** | 20.0 | Absent | Absent |
| N (perpendicular orientation) | 9.0 | 5.0[a] | **18.6** | 11.0[a] |
| B (parallel orientation) | 5.9 | −6.2 | 5.5 | −1.8 |
| B (perpendicular orientation) | 7.6 | −3.8 | 6.9 | −16.2 |

**Notes.**
[a] Peroxide product.

when comparing the reactivity of benzene towards that of azaborine, as a large variety of products, transition states and activation energies is observed when compound I is made to react with azaborine, as described in the next paragraphs.

Attack on the azaborine nitrogen atom (Fig. 2) is kinetically viable only in the doublet state and with a perpendicular orientation, yielding an azaborine peroxide product (activation energy = 9 kcal mol$^{-1}$; reaction energy 5 kcal mol$^{-1}$). With a parallel orientation, reaction is expected to be extremely slow (activation energy = 33.4 kcal mol$^{-1}$) and yields a high energy intermediate bearing an unusual interaction between the boron moiety of the substrate and one of the porphyrin nitrogens. Surprisingly, reaction in the quartet state yields (like that in the doublet state) an azaborine peroxide product, though with a

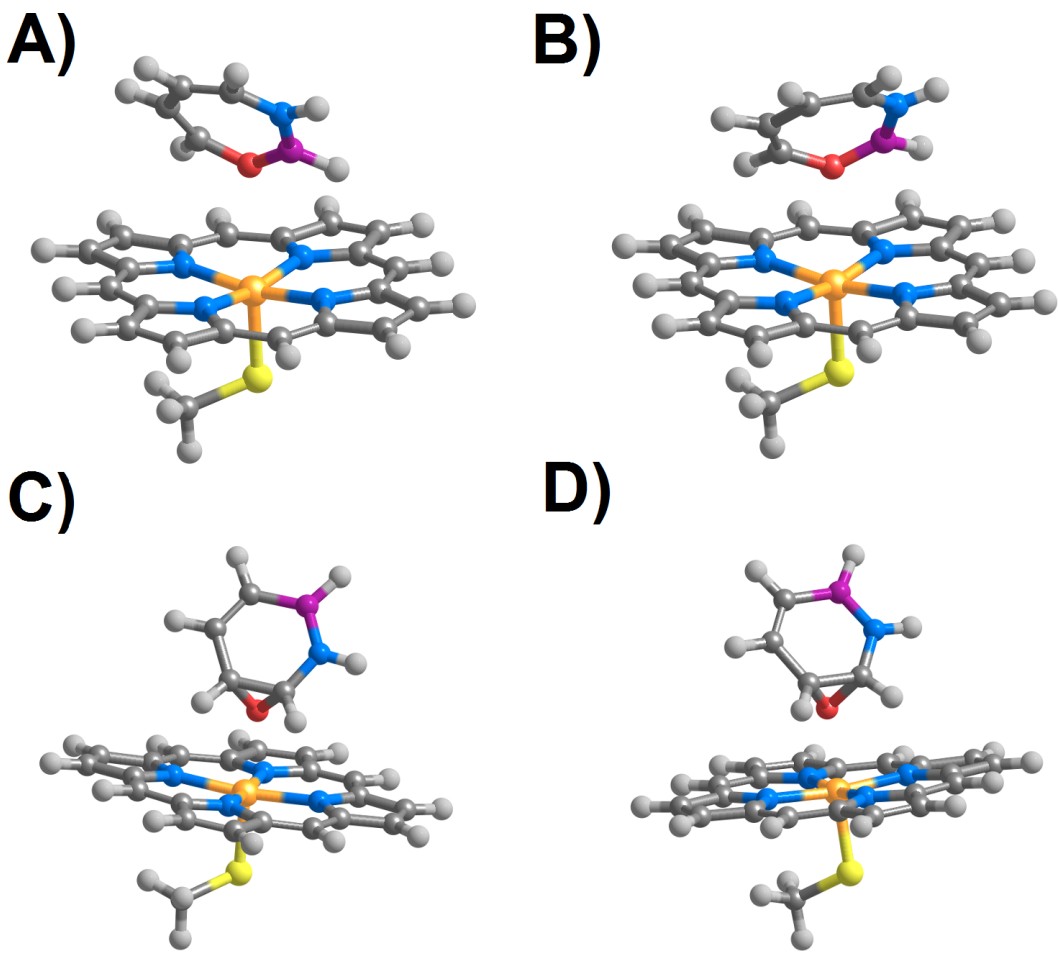

**Figure 3** Products arising from perpendicular (A and C) or parallel (B and D) attack of positions $C_3$ (A and B) and $C_5$ (C and D) in 1,2-dihydro-1,2-azaborine by compound I in the doublet ($S = 1/2$) state.

higher barrier activation energy (18.6 kcal mol$^{-1}$). In contrast, attack on the boron atom is extremely fast (with activation energies between 5.5 and 7.7 kcal mol$^{-1}$), regardless of the spin state and initial orientation of the substrate (Table 2).

Previous computational (*Silva & Ramos, 2009*) and experimental studies (*Pan, Kampf & Ashe, 2007*) ascertained that the most reactive carbon positions in azaborine towards classical electrophilic agents are its $C_3$ and $C_5$ atoms. Our computations show that the same is true regarding its reaction with the doublet state of compound I: the reaction is spontaneous by at least 47.8 kcal mol$^{-1}$ at $C_3$, and by 19 kcal mol$^{-1}$ at $C_5$. The reaction products are, however, quite different in both instances: attack on $C_3$, yields a novel heptagonal ring ($3H$-1,3,2-Oxazaborepine) containing a N–B–O–C moiety, whereas reaction in $C_5$ must overcome a 13–15 kcal mol$^{-1}$ barrier and yields epoxides over the $C_5$–$C_6$ bond. Both these products assume very similar conformations relative to the heme regardless of the initial orientation of the substrate (parallel or perpendicular) relative to the porphyrin plane (Fig. 3).

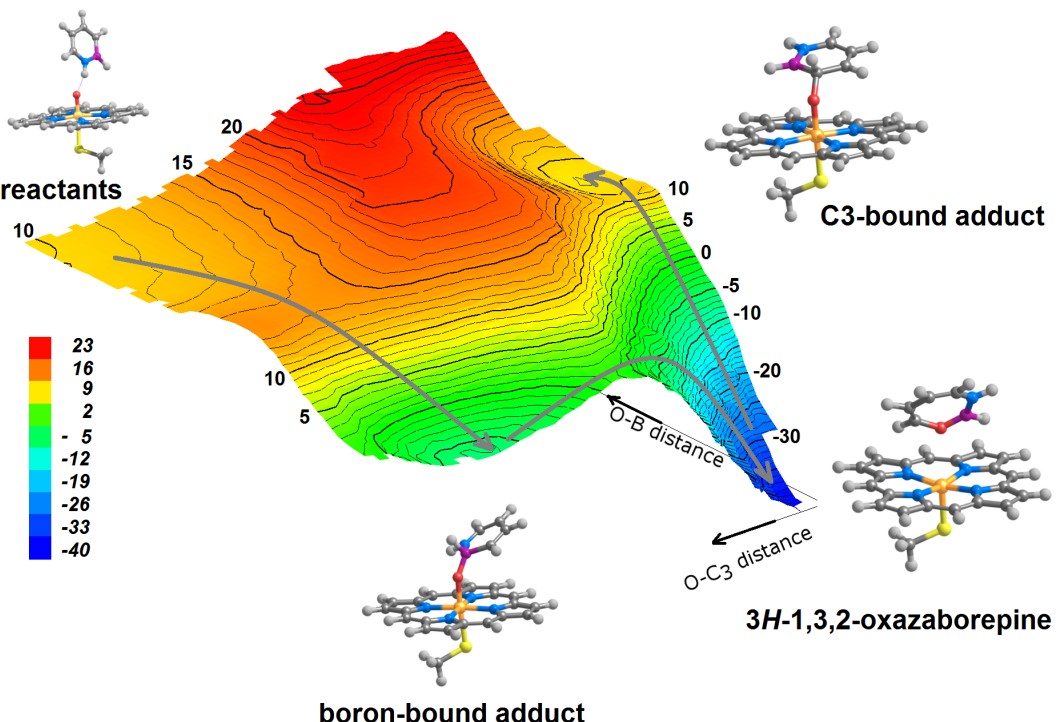

**Figure 4** **Potential energy surface obtained as B₂ and C₃ approach the reactive oxygen in compound I, computed using B3LYP with the 6-31G(d) basis set for all elements except for Fe, which used the SBKJ VDZ basis set in combination with the SBKJ pseudo-potential for the inner shells corresponding to the (1s2s2p) core of Fe.** No solvation or dispersion effects are included. Isoenergetic lines are depicted at 1 kcal mol$^{-1}$ intervals. Separated reactants with a perpendicular arrangement (corresponding to 0 kcal mol$^{-1}$) would lie far to the upper left corner of this depiction of the potential energy surface. Grey arrows show the sequence of transformations allowed as B₂/C₃ atoms approach compound I. 3H-1,3,2-oxazaborepine is only accessible after the boron-bound adduct has been formed; the C₃-bound compound I intermediate is shown to be kinetically inaccessible.

The search for a transition state for the attack on $C_3$ showed that the formation of 3$H$−1,3,2-oxazaborepine cannot occur directly from the isolated reactants, as no transition state connects this product to the reactant state: instead, 3$H$−1,3,2-oxazaborepine is formed from the boron-bound azaborine-compound I adduct, after surmounting a small barrier (Fig. 4). A second intermediate bearing a $C_3$-compound I bond was found to be a local minimum in the potential-energy surface (Fig. 4, $C_3$-bound compound I intermediate), though kinetically inaccessible due to the absence of any transition state linking it to the isolated reactants: it can only be formed (upon crossing an activation barrier above 40 kcal mol$^{-1}$) through rearrangement of the extraordinarily stable oxazaborepine.

In the quartet state, attack on $C_5$ proceeds with a barrier of 17.1 (parallel) or 18.4 kcal mol$^{-1}$ (perpendicular) and yields epoxides (like the doublet state). In contrast to the doublet state, a parallel attack of the quartet state on $C_3$ yields a $\sigma$-complex similar to that found with benzene. In the perpendicular orientation, the reactivity of the quartet state towards $C_3$ is, however, identical to that found for the doublet state.

**Table 3** **Energies (in kcal mol$^{-1}$, *vs.* the most stable reactant state) of the transition states ($^2$TS and $^4$TS) and products ($^2$product and $^4$product) for the direct attack of carbon atoms in 1,2-dihydro-1,2-azaborine by compound I.** Species preceded by $^2$ are in the doublet ($S = 1/2$) state, whereas those preceded by $^4$ are in the quartet state ($S = 3/2$). All energy values include solvation effects ($\varepsilon = 10.0$), zero-point vibrational energy and dispersion effects. Transition states with activation energies above the activation energy of the reaction of compound I towards benzene are highlighted in bold. Unless otherwise noted, all products are $\sigma$-adducts of the substrate.

|  | $^2$TS | $^2$Product | $^4$TS | $^4$Product |
|---|---|---|---|---|
| C$_3$ (parallel orientation) | n.a. | −49.2[a]/1.2 | 14.5 | 2.2 |
| C$_3$ (perpendicular orientation) | n.a. | −47.8[a]/1.3 | n.a. | −40.2[a]/1.8 |
| C$_4$ (parallel orientation) | **21.3** | 23.0 | **21.8** | 11.1 |
| C$_4$ (perpendicular orientation) | **19.5** | 10.8 | **20.5** | 9.6 |
| C$_5$ (parallel orientation) | 14.8 | −19.1[b] | **18.4** | −15.6[b] |
| C$_5$ (perpendicular orientation) | 13.2 | −18.9[b] | **17.1** | −15.4[b] |
| C$_6$ (parallel orientation) | 13.4 | 1.5 | **21.6** | −0.6 |
| C$_6$ (perpendicular orientation) | 13.1 | −2.0 | 15.2 | 7.5 |

**Notes.**
[a] Formation of 3*H*-1,3,2-oxazaborepine.
[b] Formation of a peroxide product.

The activation energies for the reactions taking place at the C$_4$-position are consistently >3 kcal mol$^{-1}$ higher than the attacks on benzene, regardless of orientation and spin state. In contrast, attacks on C$_5$ by the doublet state of compound I must surmount a lower barrier than observed for benzene, and yield very stable epoxides over the C$_5$–C$_6$ bond. The same products are observed upon attack at C$_5$ by the quartet state of compound I, though in this instance the activation barriers are 4 kcal mol$^{-1}$ above those computed for the doublet state. In spite of its negligible reactivity towards classical electrophiles (*Pan, Kampf & Ashe, 2007*; *Silva & Ramos, 2009*), the C$_6$-position in azaborine is more susceptible than benzene to attack by the doublet state of compound I in either a parallel or a perpendicular orientation. In the quartet state, the parallel orientation is noticeably less prone to react than the perpendicular orientation, in spite of yielding a more stable intermediate (Table 3).

## DISCUSSION

The computations described in this paper show that most ring positions in 1,2-dihydro-1,2-azaborine are much more reactive towards compound I than the benzene ring (for which they have been proposed as biosteres). It is therefore extremely likely that the proposed inclusion of 1,2-dihydro-1,2-azaborine in drug scaffolds will have a very detrimental effect on their ability to remain unscathed in the organism unless measures are taken to ensure that the reactive azaborine portion is sterically unable to reach the active site of P450 enzymes, or that only the very unreactive C$_4$-position is able to approach compound I.

### Funding
This research was partially carried out using computational resources acquired under project PTDC/QUI-QUI/111288/2009, funded by the Portuguese Fundação para a Ciência e

Tecnologia and FEDER through Programa Operacional Factores de Competitividade–COMPETE. The FP-ENAS Research Unit further receives some support from additional Portuguese Funds through a grant from FCT—Fundação para a Ciência e a Tecnologia (UID/Multi/04546/2013). The funders had no role in study design, data collection and analysis, decision to publish, or preparation of the manuscript.

## Grant Disclosures

The following grant information was disclosed by the author:
Portuguese Fundação para a Ciência e Tecnologia: PTDC/QUI-QUI/111288/2009.
FEDER.
FCT: UID/Multi/04546/2013.

## Competing Interests

Pedro Silva is an Academic Editor for PeerJ.

## Author Contributions

- Pedro J. Silva conceived and designed the experiments, performed the experiments, analyzed the data, wrote the paper, prepared figures and/or tables.

## Data Availability

Input and output files are available at Figshare (https://dx.doi.org/10.6084/m9.figshare.1414338).

## Supplemental Information

Supplemental information for this article can be found online at http://dx.doi.org/10.7717/peerj.2299#supplemental-information.

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
