# Peer review of "Will 1,2-dihydro-1,2-azaborine-based drugs resist metabolism by cytochrome P450 compound I?"

_PeerJ, doi:10.7717/peerj.2299_

## Round 0.1 · original submission · Minor Revisions

In addition to the reviewers, I have carefully read the paper myself. Please respond to the comments in a revised version and letter of rebuttal.

·

Basic reporting

The paper describes its results with sufficient background, with good figures. From what I can tell, appropriate raw data for this field has been provided. From what I can tell also, the paper uses one of the templates.

I think one area that could be improved is about the reaction mechanism. The author models reaction of a Compound I model (quartet or doublet) with the aza,bora-benzene species. He models steps that involve, broadly, addition of the Compound I oxygen to one of the C atoms, or the B or N. Some of these steps, however, lead to products (intermediates) that are not just addition products. E.g. species D and F in Fig. 2 involve bonding to both B and N. Species A in Fig. 3 involves insertion into the C-B bond. It would really be helpful to clarify that such species are formed in a single step from Cpd I - and to explain which calculations show this to be so.

Experimental design

The calculations reported appear to meet these criteria. The author reports slight difficulties reproducing exactly some results for benzene. I think the different basis set and the use of dispersion correction may account for this, so I'm not unduly alarmed by this.

Validity of the findings

The main finding, that these species are expected to be much more reactive than normal aromatic compounds, is well established by the work. No problem here.

Additional comments

Some general comments:

Line 17, “the very sluggish C4-position”. I don’t think a position can be called sluggish. I suggest using the more precise “less reactive”.

Line 72, Computational details: it would be good to state explicitly that the B3LYP-D3 corrections were computed as single-point corrections at B3LYP structures (if that is what was done). Also, it would be good to say exactly which version of –D3 was used.

Typo, line 97-98: “the quartet state has the same energetic barrier AS the doublet state”

Line 123, “With a parallel orientation, reaction is slow…” – this should really be stated as a hypothetical, maybe “With a parallel orientation, reaction would be slow…” or “With a parallel orientation, reaction is predicted to be slow…”.

Line 165, “was found to be thermodynamically stable”. I am not sure this wording is sensible. I think the author means that it sits in a local minimum on the potential energy surface. In this particular case, I don’t think most people would call this species “thermodynamically stable”.

·

Basic reporting

In my opinion the present paper is clearly written and is conform with the PeerJ policies

Experimental design

Although I am not sure the present study which is only based on theoretical experiments provides enough information to sustain the conclusions drawn by the author, the research question remain sound and is clearly defined.

Validity of the findings

The methodology used is acceptable and the conclusions are appropriately stated.

---

## Round 0.2 · accepted · Accept

You have taken all remarks in due consideration and the paper is accordingly improved.